# Vascular Ageing: Mechanisms, Risk Factors, and Treatment Strategies

**DOI:** 10.3390/ijms241411538

**Published:** 2023-07-16

**Authors:** Jingyuan Ya, Ulvi Bayraktutan

**Affiliations:** Academic Unit of Mental Health and Clinical Neuroscience, Nottingham University, Queen’s Medical Centre, Nottingham NG7 2UH, UK; jingyuan.ya@nottingham.ac.uk

**Keywords:** ageing, vasculature, natural senotherapeutics, senolytics, senomorphics

## Abstract

Ageing constitutes the biggest risk factor for poor health and adversely affects the integrity and function of all the cells, tissues, and organs in the human body. Vascular ageing, characterised by vascular stiffness, endothelial dysfunction, increased oxidative stress, chronic low-grade inflammation, and early-stage atherosclerosis, may trigger or exacerbate the development of age-related vascular diseases, which each year contribute to more than 3.8 million deaths in Europe alone and necessitate a better understanding of the mechanisms involved. To this end, a large number of recent preclinical and clinical studies have focused on the exponential accumulation of senescent cells in the vascular system and paid particular attention to the specific roles of senescence-associated secretory phenotype, proteostasis dysfunction, age-mediated modulation of certain microRNA (miRNAs), and the contribution of other major vascular risk factors, notably diabetes, hypertension, or smoking, to vascular ageing in the elderly. The data generated paved the way for the development of various senotherapeutic interventions, ranging from the application of synthetic or natural senolytics and senomorphics to attempt to modify lifestyle, control diet, and restrict calorie intake. However, specific guidelines, considering the severity and characteristics of vascular ageing, need to be established before widespread use of these agents. This review briefly discusses the molecular and cellular mechanisms of vascular ageing and summarises the efficacy of widely studied senotherapeutics in the context of vascular ageing.

## 1. Introduction

The world has officially entered an era of ageing. Data from the World Health Organisation indicate that the number of individuals aged ≥60 years reached 1.4 billion in 2020 and may exceed 2.1 billion by 2050 [1]. Age-related diseases (ARDs) already constitute a public health and economic issue for most countries in the world and are likely to become an even bigger issue in the not-too-distant future. Among all non-communicable diseases, vascular diseases cause much of the mortality (~44%) in the elderly [2]. Vascular events, including coronary heart disease, stroke, and heart failure, contribute to more than 3.8 million deaths in Europe alone, and 65% of these deaths occur in individuals who are 75 years of age or older [3], implying that the understanding of age-related vascular disease pathogenesis is of crucial importance for not only extending the lifespan but also improving the healthspan of elderly populations. Vascular ageing makes up the biggest pathological hallmark of many age-related vascular diseases and is characterised by arterial and capillary stiffness, endothelial dysfunction, oxidative stress stemming from an imbalance between generation and destruction of reactive oxygen species (ROS), impaired angiogenesis capacity, the presence of early signs of atherosclerosis, and low-grade chronic inflammation [4].

The mechanisms underlying vascular ageing are complex and multifactorial (Figure 1). They include the accumulation of senescent cells in the vascular system, mitochondrial dysfunction, ROS overproduction, chronic inflammation, loss of proteostasis, exhaustion and/or dysfunction of stem/progenitor cells, dysregulation of certain microRNAs (miRNAs), and extracellular matrix remodelling. The determination of the relative contributions of these mechanisms to age-related vascular disease represents an important step in the development of anti-ageing strategies aiming to improve global health and reduce healthcare-related costs. In this regard, modification of lifestyle, adoption of special dietary patterns, use of synthetic or natural therapeutic agents, and miRNA supplements have attracted much recent attention. Although a wealth of promising preclinical and clinical data has thus far been generated, additional large-scale confirmatory studies are desperately needed before approval of these agents and interventions as efficacious therapeutics to delay or modulate vascular ageing. In light of the above, this review discusses the mechanisms involved in vascular ageing, highlights the interaction between vascular ageing and other cardiovascular risk factors, and summarises the available data pertaining to the anti-vascular ageing impact of lifestyle modification, diet, and synthetic or naturally extracted components.

## 2. Mechanisms of Vascular Ageing

### 2.1. Cellular Senescence

Cellular senescence, characterised by an irreversible cell cycle arrest, represents the fundamental physiological process of organismal ageing [5]. Even though senescent cells can be detected in all organs at all ages, a substantially higher percentage of senescent cells, straying from normal function and manifesting signs of oxidative stress and inflammatory damage, are frequently observed in organs affected by ARDs compared to those from young and healthy subjects [6]. As accumulation of senescent cells is associated with a progressive loss of tissue structure and function, effective elimination of senescent cells and thus blockage of their paracrine effect have been shown to extend a healthy lifespan [7,8,9,10].

The telomere attrition theory is regarded as one of the more widely accepted theories of cellular senescence. Telomeres are repeated nucleotide sequences at each end of the chromosomes that protect the DNA strands and prevent them from fusing with other strands [11]. At each cell division, the telomeres shorten owing to the incomplete replication of the linear DNA molecules by the conventional DNA polymerases [12]. Although the shelterin complex, a group of proteins that exists at the end of telomeres, recruits a reverse transcriptase called telomerase to avoid this continuous telomeric loss by adding a telomere repeat sequence to the 3′ end of telomeres, telomeres undergo a gradual shortening as a result of continuous cell division. When telomeres become too short, a DNA damage response (DDR) is triggered, signalling the cell to undergo either senescence or apoptosis [13]. DDR involves a group of genes that sense and react to specific types of DNA damage, including specific apparatus modulating apoptosis, DNA repair, cell cycle regulation, and replication stress responses [14]. Length-independent telomere dysfunction also plays a significant role in cellular senescence. DNA-damage accumulation and DDR activation with telomeric DNA are observed in senescent cells independently of telomere attrition [15].

Exposure to different types of acute sub-lethal stresses, such as oxidative stress and genotoxic agents, triggers cellular senescence in different cell types within a relatively short period of time (days rather than months), with or without modest telomere shortening. This type of senescence is termed as stress-induced premature senescence (SIPS). Induction of SIPS is characterised by initial DNA damage and subsequent activation of DDR. Most of the DNA damage is repaired within hours of exposure to stress. However, damage at telomeric regions persists for months, suggesting that SIPS may also be regarded as a somewhat telomere-dependent event [15,16]. Despite differences in the nature of stimuli, cell cycle arrest in senescence is largely mediated by activation of either the p53/p21WAF1/CIP1 or p16INK4A/retinoblastoma protein (RB) tumour suppressor pathways [17,18]. In brief, they maintain the senescence state by evoking changes in the expression of many genes through transcriptional regulators, p53 and RB. Furthermore, the cyclin-dependent kinase inhibitor p21WAF1/CIP1 acts downstream of p53, and p16INK4A acts upstream of RB and negatively regulates cell cycle progression (Figure 2). Although cellular senescence is beneficial during embryogenesis and for wound healing, tumour suppression, and host immunity throughout adult life [19], the slow but steady accumulation of senescent cells with age overwhelms the elimination processes, triggers adverse effects in various tissues, and ultimately promotes several ARDs [6,20].

### 2.2. Impact of Cellular Senescence on the Vascular System

The accumulation of senescent endothelial cells, their progenitors (EPCs), vascular smooth muscle cells (VSMCs), and/or immunocytes represents an important mechanism in the process of vascular ageing and vascular disease. Endothelial cells and VSMCs stained positive for SA-β-galactosidase and manifesting shortened telomeres and p16/p21 activation are widely observed in atherosclerotic plaques of human arteries [21]. Foamy macrophages with higher SA-β-galactosidase activity, p16INK4A expression, and higher levels of senescence-associated secretory phenotype (SASP) factor, matrix metalloproteinase-3 (MMP3), MMP-13, interleukin-1α (IL-1α), and tumour neurosis factor-α (TNF-α) production are also commonly observed in the subendothelial space in atherosclerotic mouse models [9]. The accumulation of senescent cells in the vasculature appears to trigger atherosclerosis in both a direct and indirect fashion. Indeed, decreased expression of junction proteins and increased expression of adhesion molecules on senescent endothelial cells facilitate monocyte migration, lipoprotein uptake, and macrophage/foam cell accumulation. These and the elevated release of SASP factors, such as TNF-α, IL-6, and IL-1β, aggravate vascular inflammation and accelerate atherosclerotic plaque formation [22,23,24]. By compromising NO production and vascular relaxation and also increasing the collagen/elastin ratio, senescent cells in the vasculature contribute to the progression of ARDs, notably hypertension, and, as a consequence, raise the risk of atherosclerosis [25,26,27].

### 2.3. Endothelial Cell Senescence and Dysfunction in Vascular Ageing

The endothelium, covering the entire inner surface of all blood vessels, represents a vital organ that regulates a wide range of critical functions, including vascular tone and inflammation, through the release of a wide range of vasoactive compounds such as prostacyclin and endothelin-1 [28]. In the central nervous system, endothelial cells, the cellular component of the endothelium, also form a mechanical barrier called the blood-brain barrier (BBB) between the blood and the brain and regulate the selective passage of a variety of cells, molecules, and metabolic waste between the two compartments. Inter-endothelial cell tight junctions, formed largely by interactions between the junctional proteins occludin, claudins, zonula occludens, and adhesion molecules, constitute the backbone of the BBB. Compared to their younger counterparts, senescent endothelial cells express occludin, claudins, and zonula occludens at considerably lower levels [29,30,31]. As demonstrated with increased permeability, contact of senescent endothelial cells with young ones compromises the overall barrier integrity [32]. The loss of endothelium integrity impacts the selective permeability of the vascular wall and promotes the accumulation of oxidised low-density lipoprotein cholesterol, inflammatory cytokines, and ROS in the subendothelial space [33]. Enhanced expression of vascular cell adhesion molecule-1 (VCAM-1) and intracellular adhesion molecule-1 (ICAM-1) on senescent endothelial cell plasma membranes consequently leads to an increased adhesive and prothrombotic propensity of the endothelium [24].

As briefly mentioned above, endothelial cells regulate vascular homeostasis by synthesising and releasing a wide range of vasoconstrictors and vasodilators, as well as pro- and anti-thrombogenic factors. Nitric oxide (NO) constitutes the most important endothelium-derived agent that regulates vascular relaxation, coagulation, oxidative stress, and inflammatory responses. Due to decreased endothelial NO synthase (eNOS) activity, senescent endothelial cells generate very limited amounts of NO and therefore elicit significant impairments in all the aforementioned crucial functions and perturb vascular homeostasis [26,34]. Compared to young endothelial cells, senescent endothelial cells synthesise considerably lesser quantities of vasculoprotective acetylcholine, prostacyclin, endothelium-derived hyperpolarizing factor, and thrombomodulin and greater quantities of vasocontractile and inflammatory endothelin-1, angiotensin, ROS, reactive nitrogen species, plasminogen activator inhibitor-1 (PAI-1), thromboxane A2, and von Willebrand factor, which collectively further exacerbate the vascular damage [34,35]. The inability of senescent endothelial cells to respond to vascular endothelial growth factor (VEGF) and transforming growth factor-β (TGF-β) as effectively as their younger counterparts adversely affects angiogenesis and adds to the extent of tissue/organ damage [36,37]. Specifically, in the central nervous system, cerebral endothelial cell (CEC) senescence compromises the integrity and function of the neurovascular unit and evokes BBB disruption, neurovascular uncoupling, and neurodegeneration in addition to causing usual inflammatory, pro-coagulant, and pro-oxidant responses [38]. Here, while increased secretion of pro-inflammatory SASP factors also contributes to age-related BBB breakdown, they, along with diminished availability of low-density lipoprotein receptor-related protein-1, reduce the clearance of amyloid β from the brain [39,40]. Dysfunctional CECs with decreased NO production and impaired response to vasoactive factors perturb normal blood flow and promote chronic cerebral hypoperfusion [41,42]. BBB breakdown and chronic hypoperfusion contribute to the initiation and progression of vascular cognitive loss and Alzheimer’s disease [43,44]. Taken together, these findings indicate that the progressive accumulation of senescent endothelial cells in the vasculature may be a driving force for vascular contractility, atherogenesis, inflammation, thrombosis, coagulation, cognition loss, and impaired angiogenesis, irrespective of the tissue that they reside in.

### 2.4. Vascular Smooth Muscle Cell Senescence and Dysfunction in Vascular Ageing

VSMCs make up the main cellular component of the vascular wall (tunica media). Their roles in ageing-related vascular diseases are complicated. Indeed, while the proliferation and migration of VSMCs may be beneficial for the maintenance of plaque stability and repair of ruptured atherosclerotic plaque, their migration and accumulation in the tunica intima led to arterial lumen narrowing [45,46]. VSMCs also contribute to the stability of the extracellular matrix by secreting collagen, elastin, and MMPs [47]. An increased collagen-to-elastin ratio, a crucial marker of age-related artery stiffness, is reported in the aortas of old animals [25]. Despite possessing athero-protective and plaque-stabilising capacity and displaying a contractile phenotype in a young and healthy vascular wall [48], the VSMCs isolated from the aortas of aged rodents adopt a secretory phenotype and release a diverse range of inflammatory elements, including IL-6, C-C motif chemokine ligand-2 (CCL-2), adhesion molecule ICAM-1, and MMP-9 [49]. Once activated, MMP-9 degrades collagen and weakens atherosclerotic plaques [50]. Markedly reduced expression of the telomeric repeat-binding factor-2 (TRF2) has also been implicated in senescent VSMC-mediated plaque vulnerability [51]. It is of note here that VSMCs obtained from the fibrous cap of the atherosclerotic lesions manifest significantly shortened telomeres compared to those from disease-free tunica media, and the degree of telomere shortening correlates with the severity of atherosclerosis [52].

### 2.5. Senescence-Associated Secretory Phenotype (SASP) and Vascular Ageing

As mentioned above, despite their quiescence, senescent cells remain metabolically active and secrete a large number of soluble or insoluble cytokines, chemokines, growth factors, and MMPs that collectively make up SASP [53]. SASP has both beneficial and deleterious effects. While the former effects include wound healing and tumour suppression, the latter effects include the promotion of certain cancers, atherosclerosis, insulin resistance, endothelial dysfunction, chronic inflammatory reactions, and the induction and reinforcement of cellular senescence by paracrine activity [54]. Persistent DDR initiates the formation of SASP, whereas autocrine and paracrine effects amplify the signal and reinforce the senescent state [55]. Ataxia-telangiectasia mutated (ATM) kinase, a core component of DDR signalling, activates nuclear factor-κB (NF-κB) in response to genotoxic and oxidative stress via post-translational modifications. NF-κB, an important transcription factor, in turn regulates the expression of many genes responsible for both the innate and adaptive immune responses, including those of several pro-inflammatory factors, including IL-6, IL-8, and granulocyte macrophage colony-stimulating factor (GM-CSF), that make up SASP [56]. The suppression of the sirtuins (SIRT), a family of signalling proteins involved in metabolic regulation, also contributes to the formation and modulation of SASP. Moreover, differences in cellular origin and stimuli play pivotal roles in the expression of factors that make up SASP. For instance, while replicative senescent endothelial cells were shown to produce excessive quantities of monocyte chemotactic protein-1 (MCP-1), IL-1α, IL-6, and IL-8 [57], senescent VSMC appeared to release mostly IL-1α, IL-6, IL-8, MMP-2, and MMP-9, where the blockade of IL-1α dramatically reduced the expression of IL-6 and IL-8 [50].

### 2.6. Stem Cell Dysfunction and Exhaustion in Vascular Ageing

The impaired function and exhaustion of stem/progenitor cells may in part account for the decreased capacity of organ regeneration and damage repair during the process of ageing [58]. Through maintaining the expression and activity of telomerase and suppressing metabolic activity, stem/progenitor cells minimise the accumulation of replication-associated DNA damage cells in younger populations. In contrast, considerably higher levels of DNA damage and telomere attrition coupled with stem cell dysfunction are observed in elderly individuals [59,60]. The presence of a sufficient number of functional endothelial progenitor cells (EPCs) in circulation at all times is recognised as an important prerequisite to maintaining vascular homeostasis characterised by vascular repair and regeneration. As advanced age and major cardiovascular risk factors such as atherosclerosis, smoking, and/or type 2 diabetes increase the preponderance of senescent EPCs in circulation, they are inevitably associated with structural and functional vascular damage [61,62,63,64]. Decreases in SIRT-1 and in expression of nuclear factor-erythroid-2-related factor (Nrf2), a transcription factor required for cell cycle progression and inhibition of apoptosis and inflammation, appear also to contribute to decreased availability of EPCs and increased vascular dysfunction in aged individuals [63,65,66,67].

### 2.7. Oxidative Stress

The balance between the synthesis and activities of NO and ROS is regarded as an important determinant of vascular equilibrium. ROS, produced largely by nicotinamide adenine dinucleotide phosphate (NADPH) oxidase overactivity and mitochondrial dysfunction during the ageing process, accelerate telomere shortening, oxidise DNA bases, breakdown DNA strands, and, as a consequence, potentiate DDR activity [68,69]. These then activate a series of mechanisms leading to cellular senescence via activation of cell cycle checkpoints and mitochondrial dysfunction, which reciprocally produce more ROS and help maintain this vicious circle in an active state [70]. Air pollution, alcohol consumption, tobacco smoking, infection, and radiation constitute the common sources of exogenous ROS [71,72].

As a prominent vascular risk factor, ageing not only aggravates oxidative stress but also impairs cellular resistance to oxidative stress through attenuation of antioxidant capacity in the elderly by suppression of Nrf2, a ubiquitous transcription factor that responds to oxidative stress and promotes the genes encoding antioxidant enzymes [73,74]. In addition to its direct effects on vascular tone, the exaggerated availability of ROS compromises vascular (endothelial) function by also scavenging NO and promoting a pro-inflammatory environment [75]. In light of these findings, various natural and synthetic antioxidants, including vitamins A, C, and E, coenzyme Q10, selenium, polyphenols, melatonin, and fermented papaya preparations, have been tested as potential agents to ameliorate vascular ageing [76]. Strategies associated with lifestyle changes, such as calorie restriction and physical exercise, can also attenuate oxidative stress. Attenuation of the lifespan extension in the eNOS knockout mouse model pinpoints the significance of the eNOS/NO pathway in calorie restriction-mediated organismal benefits [77].

### 2.8. Immunosenescence and Vascular Ageing

The loss of normal innate and acquired immune function in older individuals is described as immunosenescence. Ina addition to chronological ageing, persistent exposure to viral infections can also induce premature senescence of immune cells by imposing a great antigenic volume on the immune system and substantially depleting T cells [78]. However, decreases in absolute numbers of T cells, B cells, and the naïve subset may also derive from degeneration of the thymus gland in aged individuals [79]. These decreases in the absolute number and function of immune cells during ageing may cause impairments in efferocytosis and trigger chronic low-grade inflammation, tumour development, and autoimmunity [80]. Advanced age affects the distribution of T-helper (Th) lymphocyte subtypes Th1 and Th2 in a distinct manner in that increases in Th2 lymphocyte numbers and associated pro-inflammatory responses appear to play a pivotal role in atherosclerosis and atherothrombosis [81]. The innate immune responses dominated by monocytes/macrophages also play a key role in determining the balance between the progression and regression of atherosclerotic disease [82]. Natural killer (NK) cells and cytotoxic T cells play crucial roles in the recognition and clearance of senescent cells and damaged cells [83]. Despite an increase in NK cell numbers with age, the dysfunction of NK cells, associated with reduced cytokine secretion, notably interferon-γ, and diminished target cell cytotoxicity, is regarded as one of the key phenomena in age-related impairment of immune function [84,85]. It is noteworthy here that age-related accumulation of NK cells may contribute to chronic low-grade inflammation, or inflammaging [84]. A reduction in the number of CD8+ T cells, a subset of major histocompatibility class I-restricted T cells pre-programmed for cytotoxicity, constitutes a consistent marker of immunosenescence in older adults [86]. Indeed, marked declines in the absolute number and cytotoxicity of these cells in circulation are closely associated with the insufficient clearance of senescent cells in the vascular system [87]. By accelerating cellular senescence and apoptosis, chronic low-grade inflammation may promote tissue degeneration, plaque instability, and barrier disruption and constitute a key factor for vascular remodelling [58].

### 2.9. Proteostasis and Vascular Ageing

Protein homeostasis, or proteostasis, is the process that controls proteins inside the cell to preserve the health of both the cellular proteome and the organism itself. The process involves a series of highly complex pathways that affect the fate of a protein throughout its entire existence. These include the synthesis, folding, trafficking, disaggregation, and degradation of proteins in the intracellular and extracellular matrices. The stability of proteostasis is inevitably compromised during the process of physiological ageing, where protein misfolding and autophagic dysregulation contribute to the development of ARDs. Chaperones, the ubiquitin-proteasome, and the lysosome autophagy systems constitute the three main constituents of the proteostasis network. Similar levels of heat shock protein 70 (HSP70), a seminal component of the chaperone family, were found in the cardiac tissue of aged versus young mice [88]. Even so, the long-term administration of exogenous HSP70 significantly extended the lifespan in the aged mouse model [89]. Aberrations in ubiquitin-proteasome system (UPS) regulation lead to the accumulation of misfolded proteins and, thus, abnormalities in proteasome content. Recent studies have shown a potentially reciprocal relationship between increased activation of UPS and hypertension, which may be extended to other ARDs [90,91]. The suppression of autophagy in the process of vascular ageing is well-reported and ascribed to the upregulation of mammalian target of rapamycin (mTOR) and phosphoinositide 3-kinase (PI3K) pathways and dysregulation of AMPK and SIRT pathways. Spermidine, a food supplement with an autophagy-inducing effect, has been shown to restore the expression of LC3-II, an autophagy marker, increase the expression of the autophagy machinery protein Atg3, and ameliorate vascular ageing in an aged mouse model. Spermidine has also decreased aortic pulse wave velocity, restored NO-mediated endothelium-dependent dilation, and reduced oxidative stress in the old mice compared to untreated matching controls [92]. The anti-ageing effects of calorie restriction are, in part, also associated with the promotion of autophagy [93].

The endoplasmic reticulum (ER) plays an essential role in the synthesis, folding, and structural maturation of proteins, the regulation of cellular Ca^2+^ uptake and signalling, and the production of cellular lipid. Any phenomenon that disrupts ER homeostasis creates a state commonly known as “ER stress” and triggers the unfolded protein response (UPR), which counteracts ER stress to maintain proteostasis. However, consistent UPR activation may lead to maladaptive UPR. Excessive ER stress and maladaptive UPR aggravate vascular disease, which reciprocally exacerbates ER stress [94]. For instance, bone morphogenic protein-2 is reported to promote VSMC calcification via modulation of ER stress [95]. The detection of higher quantities of ER stress markers, binding immunoglobulin protein, and C/EBP homologous protein in human atherosclerotic lesions highlights the role of ER stress in vascular pathologies [96]. Indeed, inhibition of ER stress has been shown to protect endothelial cells against H_2_O_2_-induced injury and cardiomyocytes from hypoxia/reoxygenation injury, thereby proposing ER stress as a potential target for vasculoprotective strategies [97,98].

### 2.10. MiRNAs and Vascular Ageing

MicroRNAs (miRNAs), small non-coding RNAs that participate in the regulation of physiological processes through modulation of gene expression, are increasingly recognised as important regulators of cellular senescence and vascular ageing [99,100]. Figure 3 illustrates the major miRNAs that have been implicated in vascular ageing. A study aiming to explore the differences between the expression of miRNAs in young and aged populations by screening over 800 different miRNAs has found significant downregulations in the expression of nine miRNAs, namely miR-103, miR-107, miR-128, miR-130a, miR-155, miR-24, miR-221, miR-496, and miR-1538, in aged individuals [101]. Some of these miRNAs were shown to possess vascular protective effects. For example, the upregulation of miR-107 increased the expression of tight junction protein in endothelial cells and maintained the integrity of the BBB [102]. The lentiviral delivery of miR-128 was able to prevent intimal hyperplasia in a carotid restenosis mouse model [103].

An increasing number of studies have indicated that the supplementation or inhibition of several miRNAs has anti-senescence and vasculoprotective effects [104,105,106]. The expression of miR-130a is downregulated in an ageing mouse model, and inhibition of miR-130a in young endothelial cells accelerates cellular senescence and diminishes angiogenesis. Unsurprisingly, intramuscular injection of miR-130a mimic in older mice has improved neovascularization during hindlimb ischemia [107]. Unfortunately, the enforced expression of miR-130a has also been reported to promote tumour growth in a mouse model of breast cancer [108]. Similar to miR-130a, downregulation of miRNA-199a-3p in hyperglycemic conditions led to the induction of senescence in HUVECs, where the miR-199a-3p mimic significantly decelerated this impact [109]. However, the inhibition of miR-199a-3p and miR-199a-5p in HUVECs was found to increase NO production and eNOS activity in bovine aortic endothelial cells [110]. Likewise, intramuscular injection of miR-150 mimics restored blood flow and improved functional mobility in a mouse model of hindlimb ischemia [111]. The modulatory role of miRNAs in vascular ageing and ARDs provides a new dimension for precision or personalised therapies. In this case, further pre-clinical research and clinical trials are needed to evaluate the anti-ageing, vasculoprotective, and possible adverse effects of miRNA supplements and inhibitors.

## 3. Vascular Risk Factors and Vascular Ageing

The progression of vascular ageing has a close association with several widely occurring risk factors, including type 2 diabetes, hypertension, smoking, obesity, and dyslipidemia. The accumulation of senescent cells in different organs is thought to contribute to the progression of multiple cardio-cerebral vascular risk factors, and the environmental changes caused by these risk factors reciprocally accelerate vascular ageing. Hence, a better understanding of the correlations between vascular ageing and the risk factors may culminate in the development of novel and individualised treatment strategies targeting senescent cells, SASP, or other senescence-associated dysfunction in elderly individuals as part of the therapeutic strategy aiming to control or treat vascular diseases.

### 3.1. Hypertension

Vascular ageing and hypertension represent two causally related phenomena. Indeed, hypertension is regarded as one of the most prominent ageing-related disorders. According to recent data from the National Health and Nutrition Examination Survey by the American Heart Association, over 70% of the older population (≥65 years old) and about 40% of the young adults (≤45 years old) in the US are hypertensive [112]. Observation of an elevated expression of p16INK4a and p38 MAPK, a regulator of SASP, in the ventricular artery tissue of deoxycorticosterone acetate-salt-induced hypertensive rats indicates the role of cellular senescence in the pathogenesis of hypertension [113]. Observation of oxidative stress, chronic inflammation, vascular dysfunction, vasocontractile phenotype, and endothelial permeability in individuals with advanced age confirms the causal relationship between vascular ageing and hypertension [114,115]. Dramatic reductions in both systolic and diastolic blood pressures in individuals taking quercetin, a widely investigated natural senolytic, further corroborate this relationship [116].

### 3.2. Diabetes

Recent data from the International Diabetes Federation in 2021 estimate that about 537 million adults globally suffer from diabetes, of whom about 20% are above 65 years of age and about 90% are type 2 diabetics [117]. The diabetic microenvironment predisposes the vasculature to the ageing process [118] and perturbs endothelial cell, VSMC, and EPC function through activations of growth hormone/insulin-like growth factor-Ⅰ and SIRT1/dimethylarginine dimthylaminohydrolase/asymmetric dimethylarginine pathways, which then systemically induce the process of vascular ageing [62,119]. Senescent cells can directly cause insulin resistance and aggravate the burden of diabetes through SASP containing MCP-1, insulin-like growth factor binding protein 3 (IGFBP-3), high mobility group box 1 (HMGB-1), IL-6, and PAI-1 [120,121,122]. The vicious cycle between age-related diabetes and vascular complications suggests that an effective therapeutic approach with senolytics and senomorphics may help preserve vascular homeostasis and extend a healthy lifespan.

### 3.3. Smoking

Smoking is widely regarded as a risk factor that accelerates the ageing of the respiratory, neurological, and vascular systems [123,124,125]. A cohort study looking into the correlation between smoking and cardiovascular ageing has reported that the degree of carotid plaques and vascular stiffness in current smokers is similar to those seen in their 20-year-old non-smoking counterparts [125]. The percentage of senescent and dysfunctional EPCs with impaired angiogenic function appears to be considerably higher in smokers compared to non-smokers [64]. Furthermore, endothelial cells isolated from patients who smoke go into stress-induced premature senescence at a significantly higher rate and faster pace than those isolated from non-smoking counterparts, where oxidative stress appears to play a seminal role [126]. Similar to cigarette smoking, exposure to cigarette smoke extract also induces endothelial cell senescence, apoptosis, and dysfunction via ROS and mitochondrial dysfunction, which are repressed by ubiquinol and menaquinone-7 supplements due largely to their antioxidant effects [127].

### 3.4. Virus Infection

Virus infection-related premature senescence was observed in patients with various chronic virus infections, such as Human Immunodeficiency Virus (HIV), Hepatitis B/C/D Virus (HBV/HCV/HDV), Epstein Barr Virus (EBV), Human Herpes Virus (HHV), Human Papilloma Virus (HPV), and a strain of coronavirus, i.e., SARS-CoV-2 [78,128,129,130]. In patients with HIV infection, premature senescent stem cells, endothelial cells, and leukocytes were observed. The premature senescence may be associated with the increased oxidative stress and mitochondrial dysfunction induced by HIV proteins “tat” and “nef” [131]. SARS-CoV-2 infection accelerates the process of endothelial cell senescence and evokes the release of SASP. The conditioned medium (CM) obtained from SARS-CoV-2 spike-transfected cells has been shown to induce endothelial cell senescence in a paracrine fashion, where increases in the expression or release of HMGB1, IL-1α, IL-6, and MCP-1, ROS, and adhesion molecules observed in the spike-CM-treated endothelial cells appeared to play pivotal roles [129].

### 3.5. Other Metabolic Disorders

Due to age-dependent alterations in cholesterol metabolism, hypercholesterolemia represents a major risk factor for vascular diseases in elderly [132]. Observation of key features of accelerated cellular senescence and vascular dysfunction, including shortened telomere length, increased p16 and p21 mRNA expression, and attenuated proliferation and repair capacity of endothelial cells, EPCs, and hematopoietic stem cells in a diet-induced mouse model of hypercholesterolemia, consolidates this notion. Excessive production of ROS and activation of the Notch1 pathway are among the major mechanisms that account for hypercholesterolemia-induced vascular ageing [133,134].

Hyperhomocysteinemia represents another important risk factor for vascular disease, and the mechanisms underlying hyperhomocysteinemia-induced vascular disease remain largely unknown. Even so, accumulating evidence bestows a role on oxidative stress in this process. Hyperhomocysteinemia induced in rats by a diet rich in methionine increased the expression of senescent markers SA-β-galactosidase, p53/p21, and p16 in VSMCs and elevated pulse pressure and collagen fibre deposition [135]. Similarly, hyperhomocysteinemia appeared to inactivate telomerase activity in endothelial cells and EPCs and accelerate their senescence, in which oxidative stress coupled with diminished availability of NO were instrumental [136]. PAI-1 is another important regulator of homocysteine-induced vascular ageing. Indeed, the inhibition of PAI-1 abrogates the impact of hyperhomocysteinemia on vasculature, negates reductions in the antioxidant enzyme catalase and antioxidant gene regulator Nrf2, and delays endothelial cell senescence [137].

## 4. Anti-Vascular Ageing Strategies

Anti-ageing strategies may be divided into two types according to the actual age and health status of the individuals concerned. While individuals without ARDs and major cardiovascular risk factors may benefit more from lifestyle modifications and food supplements with anti-ageing potential, elderly individuals with ARDs and/or cardiovascular risk factors may benefit from a more comprehensive strategy targeting both ageing-related mechanisms and risk factors to gain optimal benefit. Although many synthetic and natural agents with so-called anti-vascular-ageing potential have thus far been identified, changes in lifestyle such as calorie restriction and physical exercise have been put forward to delay the ageing process and improve the health and lifespan of the global population. In this context, food supplements with anti-ageing effects have also received great attention in recent years.

### 4.1. Senolytics and Senomorphics Related to Vascular Ageing

Given the crucial role of senescent cell accumulation in the pathogenesis of various ARDs such as vascular diseases, cancer, and metabolic disorders, therapeutic strategies targeting senescent cells have recently gained considerable importance. Since senescence cannot be reversed by any of the current clinical interventions, present senotherapeutics focus either on the removal of senescent cells (senolytics) or the modulation of their secretome (senomorphics).

Senolytics selectively obliterate senescent cells [138] and are divided into seven categories as per their mechanism of action: tyrosine kinase inhibitors, heat shock protein 90 (HSP90) inhibitors, B-cell lymphoma 2 (BCL-2) family inhibitors, mouse double minute 2 (MDM2) inhibitors, Forkhead Box O4 (FOXO4) inhibitors, glutaminase inhibitors, and histone deacetylase inhibitors [139]. Senomorphics modulate the function of senescent cells, especially that of SASP, and are divided into nine categories as per the elements and pathways involved in their mechanism of action: telomerase activators, sirtuin activators, mTOR inhibitors, antioxidants, NF-κB inhibitors, ATM inhibitors, Janus kinase (JAK) inhibitors, signal transducer and activator of transcription proteins (STAT) inhibitors, and p38 MAPK inhibitors [7,140]. Figure 4 summarises the current senotherapeutics. Although many senotherapeutics exert anti-ageing activity, only those known to display anti-vascular ageing are discussed in detail in this review.

As senolytics, tyrosine kinase inhibitors like dasatinib can influence many beneficial effects, ranging from tumour suppression and immune system activation to the elimination of senescent cells [141]. When combined with quercetin, a plant flavonol, dasatinib (D + Q) markedly decreases the number of senescent cells in vitro with no deleterious effect on young cells and extends the healthy lifespan in a mouse model [142]. Significant increases in NO bioavailability and reductions in aortic calcification in old mice and hypercholesterolemic mice prove the vascular protective effect of D + Q [143]. An open-label, phase 1 clinical trial consolidates these findings by showing that 3 days of oral D + Q treatment can significantly decrease senescent cell burden, including the numbers of p16INK4A+ cells, p21CIP1+ cells, and circulating SASP factors, in 55–79-year-old patients with diabetic kidney disease without noticeable adverse effects [144]. Although another clinical trial has also shown marked improvements in physical functions in 55–84-year-old patients with idiopathic pulmonary fibrosis receiving D + Q treatment, it has also reported some adverse events, including coughing, gastrointestinal discomfort, heartburn, and headache in these patients [145].

HSPs stabilise unfolded or misfolded proteins and thus give cells time to repair damaged proteins under stressful conditions such as heat shock or hypoxia. Downregulation of the anti-apoptotic PI3K/AKT pathway is implicated in the senolytic function of inhibitors targeting HSP90 at cellular and organismal levels [146]. Glutaminase is a mitochondrial enzyme that metabolises glutamine to glutamate and ammonia [147]. There are two isoenzymes: kidney type (glutaminase-1) and liver type (glutaminase-2). These two isoenzymes are expressed differently in different tissues and often display contrasting effects, notably on cancer growth [148]. Glutaminase inhibitors have been shown to rejuvenate human skin via clearance of senescent cells in a study using a mouse/human chimeric model where prominent decreases in various SASP elements, such as MMP-3, MMP-9, IL-1α, IL-6, and TNF-α, were also observed [149]. Selective targeting of glutaminase 1 in old mice has replicated these results and led to selective elimination of senescent cells and amelioration of age-associated organ dysfunction [150].

Sirtuins are a family of signalling proteins involved in metabolic regulation. They modulate the activity of their targets by removing covalently attached acetyl groups and are implicated in numerous biological pathways. Hence, they are recognised as highly promising therapeutic targets. There are seven sirtuins in mammals: SIRT1–7. A growing body of evidence implicates that sirtuins function is crucial to vascular ageing [151]. Activation of SIRT1 in human EPCs with MHY2233 protects cells against both replicative and oxidative stress-induced senescence and maintains normal function [35]. Resveratrol, a type of natural phenol and a phytoalexin found in red grapes, berries, peanuts, and tomatoes, also exerts its multiple anti-ageing effects, including its antioxidant and anti-inflammatory roles, through the activation of SIRT1, AMP-activated protein kinase (AMPK), and Nrf2 pathways, eNOS phosphorylation, and concomitant inhibition of mTOR and NF-κB pathways [152,153,154,155]. Oral administration of resveratrol supplements has been found to reduce the expression or availability of senescence biomarkers and preserve arterial function against the effects of vascular ageing in aged mice and rodent models. Animals that received resveratrol had a lower level of media thickness in the aorta, less inflammation, fibrosis, and oxidative stress, and displayed increased aerobic capacity and exercise tolerance [155,156]. The findings of a double-blind, randomised, placebo-controlled clinical trial conducted with type 2 diabetes patients receiving 100 mg/day resveratrol for 12 weeks supported these findings and revealed significantly lower systolic BP, cardio-ankle vascular index, oxidative stress, and arterial stiffness compared to the placebo group [157]. Similarly, another parallel-design, double-blind, randomised clinical trial revealed that older patients with peripheral artery disease on a lower dose (125 mg/day) resveratrol supplement achieved greater 6-min walk improvement than the ones in the placebo group or on a higher dose (500 mg/day) resveratrol supplement [158].

As a widely investigated senomorphic, the mTOR pathway inhibitor rapamycin has been shown to extend both the median and maximal lifespan in mouse models. The mTOR pathway is a central regulator of mammalian metabolism and physiology, with important roles in cell growth and proliferation, angiogenesis, cell survival via DNA repair, and suppression of autophagy, as well as apoptosis and bioenergetics through augmenting metabolism and nutrient availability. Indeed, increasing evidence attributes the anti-ageing effects of rapamycin to its ability to regulate translation and induce autophagy. However, rapamycin has also been shown to negate the deleterious effects of oxidative stress on vascular tone through, in part, maintenance of stem cell function and anti-inflammatory effects [159,160]. Despite these experimental findings, due to its substantial adverse effects, including suppression of the immune system, anaemia, and renal toxicity, rapamycin has not been tested in clinical settings other than transplant surgery and renal cell carcinoma [161,162].

Several epidemiological studies have demonstrated that metformin reduces the incidence of various ARDs and all-cause mortality in both diabetic and non-diabetic patients. Findings of a randomised, double-blind, placebo-controlled, crossover trial exhibit that metformin regulates both metabolic and non-metabolic pathways associated with ageing in the elderly, providing good evidence for the anti-ageing impact of metformin [163]. Preclinical data showing that metformin extends the lifespan of Caenorhabditis elegans and mice further supports this notion. Increases in AMP-activated protein kinase activity and concurrent activation of Nrf2-dependent antioxidant responses leading to suppression of oxidative stress and inflammatory responses may be involved in the beneficial effects of metformin [10].

Considering that senescent cells mediate both autocrine and paracrine effects through SASP, modulation of SASP is considered an important therapeutic strategy. In this regard, p38MAPK, a major transducer of stress stimuli that is activated in response to diverse cellular and environmental stresses such as DNA damage, oxidative stress, and pro-inflammatory cytokines, attracts much attention. P38MAPK mediates various events, including cell proliferation, cell cycle progression, and apoptosis [164]. Inhibition of p38MAPK suppresses the secretion of many SASP factors such as IL-1α, IL-1β, IL-6, IL-8, and GM-CSF and delays senescence in different cell lines, e.g., fibroblasts and bone primary cells subjected to repetitive passage, lipopolysaccharides, or oncogenes [165,166,167]. Doramapimod, a p38MAPK inhibitor, has proven to be particularly efficacious in delaying senescence and upregulating tight junction protein expressions in brain microvascular endothelial cells to restore BBB integrity [30]. SB203580, another p38MAPK inhibitor, has been shown to alleviate vascular stiffness in Angiotensin Ⅱ-infused hypertensive mice by normalising the elastin structure [168].

### 4.2. Lifestyle Modification

Lifestyle modification involves changing long-term habits, typically of eating or physical activity, and maintaining the newly acquired behaviour for months or years. Lifestyle modification may be effective in treating a range of diseases and pathological phenomena, including obesity and age-related cellular senescence.

#### 4.2.1. Exercise

Being physically active slows the ageing process and helps people live healthier and more vigorous lives. Reduced bioavailability of NO and chronic inflammation triggered by oxidative stress are among the key mechanisms that induce endothelial dysfunction and vascular disease and, as a consequence, accelerate vascular ageing [27]. Substantial increases in antioxidant glutathione and decreases in pro-oxidant superoxide anion and acute phase reactant C-reactive protein levels in human subjects immediately after physical exercise confirm the presence of a link between exercise and oxidative stress [169,170]. Amelioration of arterial remodelling through maintenance of NO-dependent vascular relaxation and reduction of overall inflammation by long-term exercise in individuals with cardiovascular disease corroborate this link further and highlight the importance of regular exercise in the delay of vascular disease and ageing [171]. Contrary to the increases in older populations and young sedentary adults, observation of significantly lower levels of senescence biomarkers p53, p16, and p21 in endothelial cells across different age groups of habitually exercising adults confirms this notion [172]. In addition to its effect on ageing, even a relatively short period of aerobic exercise (12 weeks) also improves both physical and mental performance in older adults [173]. This is further supported by the results of a randomised clinical trial (RCT) demonstrating that 16 weeks of endurance exercise training peaks exercise oxygen consumption by enhancing skeletal muscle perfusion and oxygen utilisation without improving endothelial function or arterial stiffness in elderly patients with heart failure [174]. Another RCT study focusing on the beneficial effects of exercise in prehypertensive or hypertensive patients suggests that all types of physical exercise, including aerobic exercise training, resistance training, and combined training, significantly reduce both systolic and diastolic blood pressures and improve endothelial function, as evidenced by increased artery flow-mediated dilation [175].

#### 4.2.2. Calorie Restriction

A guided reduction in food intake or calorie restriction constitutes another lifestyle modification-mediated anti-ageing intervention. By diminishing both the age-related decline in physiological fitness and the risk of ARD development, calorie restriction extends the healthy lifespan [176,177] in that increases in telomerase activity and SIRT1 and SIRT6 expressions appear to play critical roles [178,179]. Results of a two-year clinical trial conducted with healthy and non-obese individuals show that calorie restriction may also improve overall health and wellbeing by significantly attenuating oxidative stress [180]. Elevations in the expression of chaperones and cellular autophagy markers in healthy volunteers, crucial factors for proteostasis stability, may also account for the beneficial effects of calorie restriction [181].

### 4.3. Dietary Modulation

Dietary modulation, including alteration of food composition and intake of food supplements, may be efficacious in controlling vascular risk factors and vascular ageing [182,183]. Healthy dietary patterns can not only provide better control of several vascular risk factors but can also independently delay vascular ageing and vascular disease development in healthy elderly populations. The dietary guidelines of the American Heart Association recommend the consumption of whole grain foods and products, healthy sources of protein (mostly plants, fish, seafood, low-fat dairy, and lean, unprocessed meat if desired), and liquid plant oils for the improvement of cardiovascular health. It also recommends controlling energy intake to maintain a healthy body weight and minimising the intake of added sugar, salt, ultra-processed foods, and alcohol [184]. In line with these general principles, several existing dietary patterns are known to have vasculoprotective effects. The traditional Mediterranean diet (MedDiet), a diet rich in vegetables, fruits, olive oil, cereals, legumes, fish, and seafood, stands out as an important dietary pattern in this regard. Indeed, MedDiet has been shown to reduce oxidative stress and systemic inflammation and augment circulating EPC numbers, which in turn are associated with the reduced prevalence of ageing-related risk factors and prolonged longevity in the elderly population in multiple trials [185,186,187,188]. Other dietary approaches, such as those aiming to stop hypertension (DASH), reduce serum cholesterol (the portfolio diet), and the Okinawan diet, which is low in calories and fat but high in carbohydrates, have also been effective in prolonging health span [182].

As discussed above, diets based on low calorie intake, in general, positively correlate with healthy ageing and lifespan. MedDiet appears to be an exception to this rule [189]. The results of a RCT comparing the cardiovascular disease preventive effects of MedDiet and a low-fat diet with no restriction on calorie intake indicate the superiority of MedDiet in preventing major cardiovascular events such as myocardial infarction, stroke, and peripheral artery disease [190]. In addition to being a low-cholesterol and low-energy-dense diet, the abundance of specific bioactive molecules, such as oleic acid and phenolic components from olive oil, omega-3 fatty acids from fish, and resveratrol from grapes, helps explain the vasculoprotective effects of MedDiet. Similarly, the traditional Okinawan diet, containing huge quantities of vegetables, soybean products, herbs, and spices alongside moderate amounts of fish and lean meat, has many vasculoprotective effects, which are in part attributed to significant increases in circulating EPC counts even after 2 weeks of dietary intervention [191]. It is anticipated that successful identification of the components in these dietary patterns can eventually lead to the development of food supplements with vascular protective and anti-ageing effects.

### 4.4. Food and Food Supplements with Anti-Vascular Ageing Effects

#### 4.4.1. Olive Oil and Its Extracts

Since individuals on MedDiet supplemented with extra-virgin olive oil develop lower incidences of stroke, myocardial infarction, and vascular dysfunction-related death compared to those who consume nuts and a reduced-fat diet as part of their MedDiet [186], it is reasonable to suggest that virgin olive oil phenolic extract (OOPE) has significant anti-ageing, vascular protective, and neuroprotective effects. The discovery of the anti-inflammatory, lipid peroxidation inhibitory, and glutathione restoration activities of OOPE supports this hypothesis [192,193]. Mitigation of oxidative stress-induced nuclear DNA damage in HeLa cells and peripheral blood mononuclear cells by OOPE also demonstrates its antioxidant effects [194,195]. Preclinical studies showing a markedly lower level of DNA damage and telomere attrition in ageing animals fed with olive oil rather than other fat sources corroborate this finding [196,197]. A cohort study with 3349 middle-aged to elderly participants from the Mediterranean countries investigating the anti-ageing effects of olive oil and other dietary fats has revealed that exclusive consumption of olive oil is better associated with the “successful ageing index”, a quantifiable score to describe the psychological, clinical, and social ability of the elderly individuals [198].

#### 4.4.2. Long-Chain Omega-3 Fatty Acids

Omega-3 fatty acids (O3FAs) are polyunsaturated fatty acids. They are important constituents of animal lipid metabolism, and they play a crucial role in the human diet. A-linolenic acid (ALA), eicosapentaenoic acid (EPA), and docosahexaenoic acid (DHA) make up the three O3FAs involved in human physiology. While ALA is found in plants, DHA and EPA are found in fish. Since humans cannot synthesise ALA, they need to obtain it through diet. Although humans can form EPA and DHA using ALA, the capacity to form longer O3FAs is impaired during ageing [199]. Due to the high consumption of fish, traditional MedDiet is considered a long-chain O3FA-rich diet. A double-blind RCT study exploring the effects of DHA, EPA, or fish oil (containing EPA and DHA in a ratio of 2:1) supplements on vascular inflammation and atherosclerosis has reported that supplements of O3FAs significantly increased the EPA and DHA levels in the blood without elevating cholesterol levels. The subsequent in vitro and in vivo experiments to this trial demonstrated that supplementation with EPA significantly reduced the expression of SASP factors, IL-1β, CCL2, ICAM-1, and vascular cell adhesion molecule-1 (VCAM-1) in endothelial cells exposed to TNF-α and in the vascular wall of mice fed an atherogenic diet [200]. Another RCT determined that supplements with fish oil (containing EPA and DHA) in middle-aged individuals significantly decreased the markers of aortic stiffness, aortic pulse pressure, and aortic augmentation pressure [201]. In addition to its physical effects, decreased availability of DHA in the ageing brain is also associated with cognitive decline [202]. O3FA supplements have been shown to raise the DHA level in the brains of aged animals and improve recognition memory [203].

#### 4.4.3. Curcumin

Curcumin is a phenolic component extracted from the roots of Curcuma longa, which is usually used for seasoning food and as part of traditional medicine in Asia. The capacity of curcumin to act as a prominent antioxidant through activation of the Nrf2 pathway and suppress inflammation through regulation of the NF-κB pathway has been proposed to account for its anti-ageing effects [204]. However, a RCT studying the impact of 12-week curcumin supplementation in middle-aged and elderly individuals has reported improvements in resistance and conduit artery endothelial function through an elevation in NO bioavailability without affecting the level of antioxidants and inflammatory markers in the circulation [205].

#### 4.4.4. Resveratrol

As discussed above, low doses of resveratrol supplements revealed both anti-ageing and vascular protective effects in both translational and clinical studies. However, the data regarding the optimal dose and possible adverse effects of resveratrol need further clarification before it can safely be used as a pharmaceutical supplement in the general population. Under these circumstances, increasing the intake of foods rich in resveratrol, such as grapes, blueberries, raspberries, and peanuts, remains a safe and beneficial option [206].

## 5. Discussion and Future Perspectives

The elucidation of various mechanisms involved in physiological and pathological vascular ageing has led to the discovery of a series of agents that may be utilised to delay the ageing process and improve lifelong health and wellbeing [207]. It is possible that some of the mechanisms outlined in the review for vascular ageing can also play crucial roles in the physiological and premature ageing processes of other systems. Similarly, the therapeutic strategies discussed may be equally efficacious to delay or control the ageing process of other systems too.

Individuals who present a significant loss of arterial elasticity and reduced arterial compliance due to accelerated mechanical and structural changes in the vascular wall are thought to have premature vascular ageing compared to the general population of similar age. Vascular ageing is a chronic, systemic, progressive process that affects everybody, although its characteristics and severity are dictated by the individual’s chronological age, genetics, the presence of risk factors, and their lifestyle choices. For example, physical exercise produces better anti-ageing and vascular protective effects in patients with hypertension than in those with heart failure [174,175], while O3FA supplements may provide superior benefits for elder individuals with dyslipidemia. Despite an increased understanding of mechanisms associated with vascular ageing, a series of major challenges continue to hinder person-specific interventions. Firstly, the absence of specific, standardised, and easily detectable biomarkers makes it extremely difficult to detect and monitor the status of vascular ageing in physiological and pathological settings. Secondly, the presently scant data on the possible side effects of anti-ageing interventions, especially in the healthy elderly, necessitates the performance of well-designed comprehensive studies to assess the feasibility, safety, and efficacy of treatments with different anti-ageing agents in the elderly with and without overt vascular diseases. Finally, the impact of lifestyle modifications and food supplements as well as their interaction with so-called pharmacological interventions deserve special attention.

## Figures and Tables

**Figure 1 ijms-24-11538-f001:**
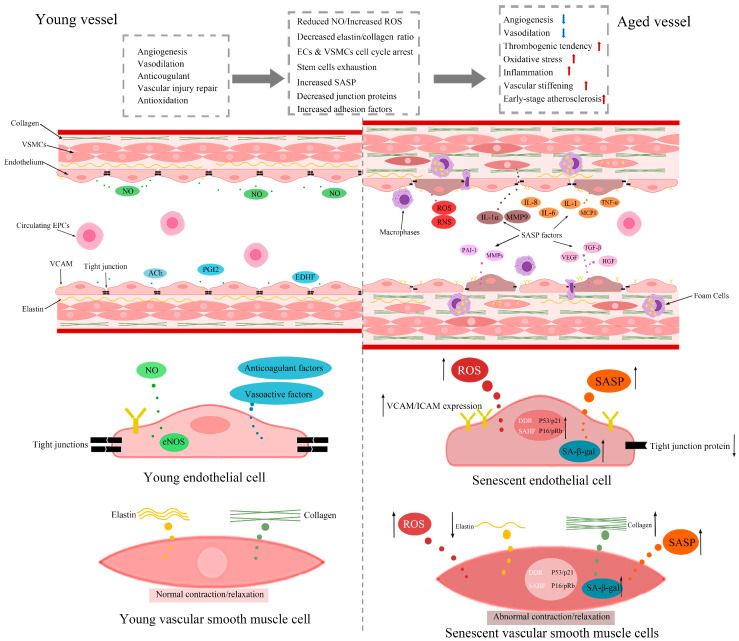
Mechanisms of vascular ageing. The presence of young and functional endothelial cells, vascular smooth muscle cells (VSMCs), and endothelial progenitor cells (EPCs) ensures that angiogenesis, vascular tone, coagulation, oxidative stress, vascular repair, and permeability are closely monitored in young vasculature at all times. To this end, young endothelial cells express high levels of tight junction proteins and low levels of adhesion molecules and secrete various vasoactive compounds with anti-inflammatory and vascular relaxant capacity, including prostaglandin (PGI2), acetylcholine (Ach), endothelium-derived hyperpolarizing factor (EDHF), and notably the most prominent endogenous vasodilator, nitric oxide (NO), through endothelial-type NO synthase (eNOS) activity. Young VSMCs, on the other hand, secrete elastin and collagen in a stable ratio to help maintain normal vascular tenacity, pliability, and contractility. In contrast, decreases in NO bioavailability, tight junction protein expression, endothelial cell proliferation, and elastin-to-collagen ratio, as well as concurrent increases in reactive oxygen species (ROS) production and adhesion molecule expression in aged vessels, lead to impaired angiogenesis and vasodilation, oxidative stress, inflammation, thrombogenesis, and vascular stiffening. Diminished availability or function of senescent EPCs and the secretion of senescence-associated secretory phenotype (SASP) comprising matrix metalloproteinases (MMPs), extracellular matrix components, growth factors, cytokines, etc. also contribute to the dysfunction of aged vessels. VCAM: vascular cell adhesion molecule 1; ICAM: intracellular adhesion molecule 1; RNS: reactive nitrogen species; PAI-1: plasminogen activator inhibitor 1; IL: interleukin; MCP1: monocyte chemoattractant protein 1; TNF: tumour necrosis factor; TGF: transforming growth factor; VEGF: vascular endothelial growth factor; HGF: hepatocyte growth factor; DDR: DNA damage response; SAHF: senescence-associated heterochromatin foci.

**Figure 2 ijms-24-11538-f002:**
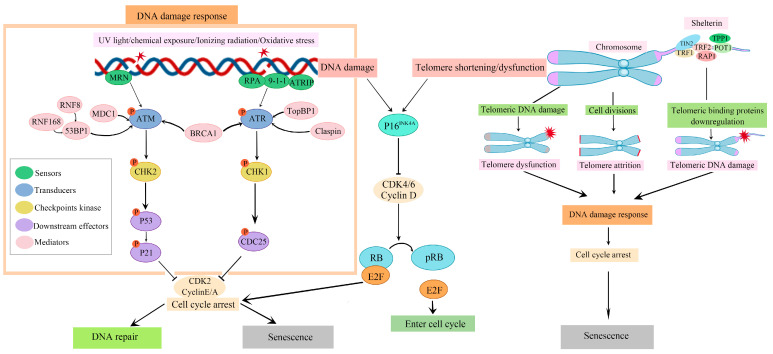
Triggers and general signalling pathways of cellular senescence. Telomere attrition, telomeric DNA damage, shelterin protein downregulation, and damages accumulated on non-telomeric DNA from other stimuli collectively trigger the DNA damage response (DDR) and p16/pRb pathway, resulting in either temporary cell cycle arrest for the DNA repair process or permanent cell cycle arrest, known as cellular senescence. DDR involves a series of processes. While the MRN (MRE11-RAD50-NBS1) complex, accompanied by activation of ataxia telangiectasia mutated (ATM)/Cell Cycle Checkpoint kinase-2 (CHK2), is implicated in DNA double-strand breaks, the 9-1-1 (RAD9-RAD1-HUS1) complex, leading to activation of ATM and rad3-related (ATR)/CHK1, is implicated in DNA single-strand breaks. Once activated, checkpoint kinases CHK1 and CHK2 trigger cell cycle arrest by inhibiting cyclin-dependent kinase 2 (CDK2) activation through the p53/p21 and CDC25 pathways. The accumulation of telomeric and non-telomeric DNA damage induces the expression of p16INK4A. The P16 protein inhibits CDK 4/6, thereby suppressing Rb phosphorylation and the release of E2F translation factor, which is crucial for cell cycle progression. RNF: ring finger containing nuclear factor; 53BP1: P53-binding protein 1; BRCA1: breast cancer type 1; RPA: replication protein A; ATRIP: ATR-interacting protein; TopBP1: DNA Topoisomerase II Binding protein 1; CDC: cell division cycle; TIN: telomeric repeat binding factor 1/2-interacting nuclear factor 2; TRF: telomeric repeat binding factor; RAP1: repressor/activator protein 1; POT1: protection of telomeres 1; TPP1: POT1-interacting protein 1.

**Figure 3 ijms-24-11538-f003:**
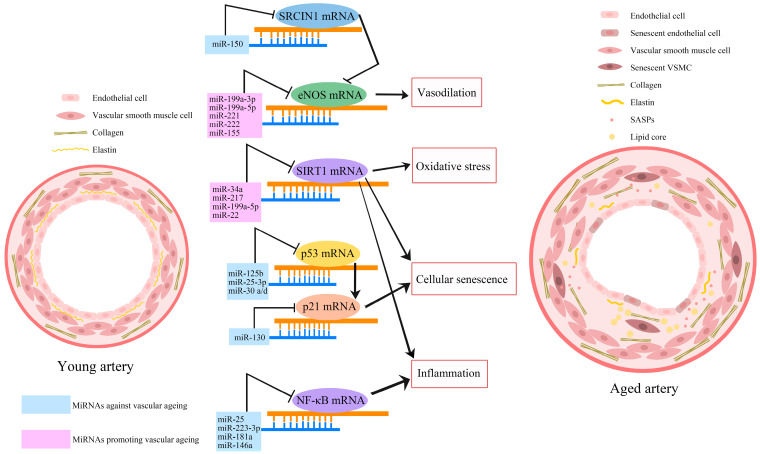
MiRNAs modulate vascular ageing. MiRNAs regulate the process of senescence by modulating gene expression. Some miRNAs promote vascular ageing (pink rectangles), while others antagonise it (blue rectangles) through their specific effects on mRNAs of SRC kinase signalling inhibitor 1 (SRCIN1), endothelial nitric oxide synthase (eNOS), sirtuin 1 (SIRT1), transcription factor nuclear factor-kappa B (NF-κB), p53, and p21, which regulate inflammation, oxidative stress, vascular tone, and vascular cell senescence. The accumulation of senescent cells in aged arteries leads to the acquisition of the senescence-associated secretory phenotype (SASP).

**Figure 4 ijms-24-11538-f004:**
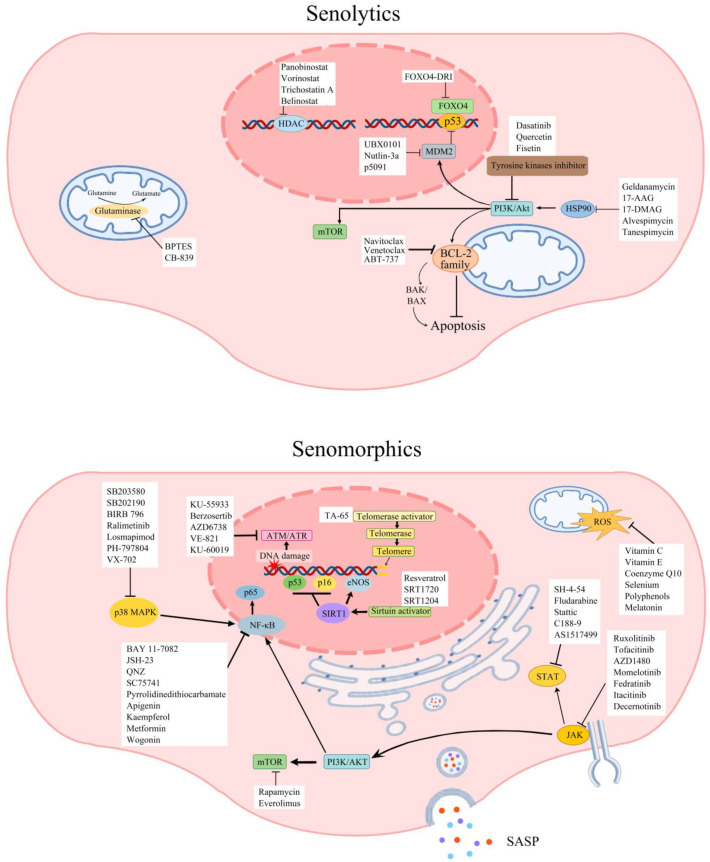
Types of currently known senolytics and senomorphics and their mechanisms of action. Several signal transduction pathways and senescence-associated secretory phenotypes are involved in the actions of senolytics and senomorphics. Currently known senolytics are categorised into seven groups: tyrosine kinase inhibitors, heat shock protein 90 (HSP 90) inhibitors, B-cell lymphoma-2 (BCL-2) family inhibitors, mouse double minute 2 (MDM2) inhibitors, Forkhead Box O4 (FOXO4) inhibitors, glutaminase inhibitors, and histone deacetylase (HDAC) inhibitors. Currently known senomorphics are categorised into nine groups: telomerase activators, sirtuin (SIRT) activators, mammalian target of rapamycin (mTOR) inhibitors, antioxidants, nuclear factor-kappa B (NF-κB) inhibitors, ataxia telangiectasia mutated (ATM) inhibitors, Janus kinase (JAK) inhibitors, signal transducer and activator of transcription proteins (STAT) inhibitors, and p38 mitogen-activated protein kinase (MAPK) inhibitors. A series of downstream signalling pathways, enzymes, and environmental changes such as phosphatidylinositol 3-kinase/protein kinase B (PI3K/Akt), senescence-associated secretory phenotype (SASP), endothelial nitric oxide synthase (eNOS), ATM and RAD3-related protein (ATR), and oxidative stress accompanied by excessive release of reactive oxygen species (ROS) play pivotal roles in the activity of senotherapeutics.

## Data Availability

No new data were created or analysed in this study. Data sharing is not applicable to this article.

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
