# Peer review of "Vascular Ageing: Mechanisms, Risk Factors, and Treatment Strategies"

_ijms, 2023, doi:10.3390/ijms241411538_

Round 1

Reviewer 1 Report

The authors did a great job explaining the mechanisms, risk factors, and treatment strategies for vascular aging. Given the rapid increase in aging population, the review on vascular aging is highly important. I have following questions and suggestions:

1.     One of my major concerns is almost all the figures are very detailed (which is a good thing) but with very little or no explanation in the text or figure legend. This can make the readers lost if they are not familiar with the illustrated pathways. Therefore, it will be beneficial to provide a brief description of all the pathways illustrated in the figures. 

2.     Figure 1, explain the difference between the grey and red arrows. Also, the top part is very confusing to make a concrete conclusion without any explanation in the figure legend. 

3.     Section 2.2, provide a brief link between atherosclerosis and vascular aging. 

4.     Ref required for lines 185-188, 209-211. 

5.     What is DDR. I could not find the introduction to DDR. 

6.     Provide a brief introduction to the role of Nrf2 in line 254. 

7.     Elaborate the link between vascular aging (or vascular diseases) with immunosenscence. 

8.     Figure 3 shows the role of miR-221,222,155 in regulation of eNOS but in text you are talking about miR199 and NO. 

9.     Elaborate on the link between exercise and oxidative stress and how it beneficial in vascular aging. 

10.  Include AHA dietary guidelines on cardiovascular diseases in section 4.3. 

Minor editing. 

Reviewer 2 Report

In this paper the authors make a deep review of the molecular and cellular mechanisms underlying vascular ageing characterized by vascular stiffness, endothelial dysfunction, increased oxidative stress, among others, and their association to several age-related vascular diseases and treatments. Although deep, I miss the discussion of some points that might improve the review:

     1) What does it know about physiological ageing and premature ageing in terms of vascular ageing? All the mechanism described in this review could be translated to those diseases associated with premature ageing? Both senolytics and senomorphics strategies might be used in this field?

     2) In the section 2.3, the authors should include some lines about the relationship between endothelial dysfunction and ageing with some neurological diseases such as Alzheimer’s

    3) In the section 2.9, it would be interesting to study the current state of the endoplasmic reticulum stress inhibitors in relation within the protein homeostasis.

Minor points:

-        In the section 2.1, authors define cellular senescence as “an irreversible cell cycle arrest after a number of passages, represents the fundamental physiological process”. However, the relationship between in-vitro approaches (number of passages) with physiological processes are discussable. Please, rephrase the sentence in a more general terms.

-        In lines 91-93 there is some text in italics font. Is this something related and relevant with the text or only an oversight?

Round 2

Reviewer 1 Report

Authors addresses my concerns. 

none